# Effects of Ferroptosis on the Metabolome in Cardiac Cells: The Role of Glutaminolysis

**DOI:** 10.3390/antiox11020278

**Published:** 2022-01-29

**Authors:** Keishla M. Rodríguez-Graciani, Xavier R. Chapa-Dubocq, Esteban J. Ayala-Arroyo, Ivana Chaves-Negrón, Sehwan Jang, Nataliya Chorna, Taber S. Maskrey, Peter Wipf, Sabzali Javadov

**Affiliations:** 1Department of Physiology, School of Medicine, University of Puerto Rico, San Juan, PR 00936, USA; keishla.rodriguez20@upr.edu (K.M.R.-G.); xavier.chapa@upr.edu (X.R.C.-D.); esteban.ayala@upr.edu (E.J.A.-A.); ivana.chaves@upr.edu (I.C.-N.); sehwan.jang@upr.edu (S.J.); 2Department of Biochemistry, School of Medicine, University of Puerto Rico, San Juan, PR 00936, USA; nataliya.chorna@upr.edu; 3Department of Chemistry, University of Pittsburgh, Pittsburgh, PA 15260, USA; taber.maskrey@pitt.edu (T.S.M.); pwipf@pitt.edu (P.W.)

**Keywords:** ferroptosis, cardiomyocytes, metabolome, mitochondria, glutaminolysis, anti-ferroptotic compounds, ferrostatin-1, XJB-5-131, TSM-1005-44

## Abstract

Ferroptosis is a novel iron-dependent regulated cell death mechanism that affects cell metabolism; however, a detailed metabolomic analysis of ferroptotic cells is not yet available. Here, we elucidated the metabolome of H9c2 cardioblasts by gas chromatography-mass spectrometry during ferroptosis induced by RSL3, a GPX4 inhibitor, in the presence of ferrostatin-1 (a ferroptosis inhibitor), XJB-5-131 (a mitochondrial-targeted ROS scavenger), or TSM-1005-44 (a newly developed cellular ROS scavenger). Results demonstrated that RSL3 decreased the levels of amino acids involved in glutathione synthesis more than two-fold. In contrast, saturated fatty acids levels were markedly increased in RSL3-challenged cells, with no effects on unsaturated fatty acids. RSL3 significantly altered the levels of mitochondrial tricarboxylic acid cycle intermediates; isocitrate and 2-oxoglutarate were found to increase, whereas succinate was significantly decreased in RSL3-challenged cells. Ferrostatin-1, XJB-5-131, and TSM-1005-44 prevented RSL3-induced cell death and conserved the metabolomic profile of the cells. Since 2-oxoglutarate is involved in the regulation of ferroptosis, particularly through glutamine metabolism, we further assessed the role of glutaminolysis in ferroptosis in H9c2 cardioblasts. Genetic silencing of GLS1, which encodes the K-type mitochondrial glutaminase (glutaminase C), protected against ferroptosis in the early stage. In conclusion, our study demonstrates that RSL3-induced ferroptosis impairs the metabolome of H9c2 cardioblasts.

## 1. Introduction

Ferroptosis is a newly discovered iron-dependent non-apoptotic programmed cell death pathway. It is characterized by the accumulation of oxidized phospholipids due to the excessive oxygenation of polyunsaturated fatty acid (FA) residues of phospholipids by lipoxygenases and the limited capability of glutathione peroxidase 4 (GPX4) to neutralize oxidized phospholipids [1,2,3,4]. Mitochondria are the main source of reactive oxygen species (ROS); electron transport chain (ETC) complexes, monoamine oxidase, α-ketoglutarate dehydrogenase, NADPH oxidase 4, among others, produce superoxide (O_2_^•−^), which is reduced to hydrogen peroxide (H_2_O_2_) upon interacting with superoxide dismutase [5]. Normally, catalase and H_2_O_2_-specific glutathione peroxidases convert H_2_O_2_ into H_2_O; however, under pathological stimuli, free redox-active iron becomes available in the cytosol. Redox-active iron through the Fenton reaction further increases the accumulation of ROS and activates lipoxygenases, particularly 15-lipoxygenases. The latter stimulates oxidation of free polyunsaturated FA esterified into phospholipids followed by membrane damage, thus leading to ferroptosis [1,2,6].

Although there are eight isoforms of GPX enzymes, early studies have identified the inhibition of the GPX4 isoform to be the main contributor of ferroptosis due to its ability to reduce oxidized phospholipids, such as phosphatidylethanolamine hydroperoxides, FA hydroperoxides, and cholesterol, among others [4]. GPX4 reduces the hydroperoxy-phospholipids to stable hydroxy-phospholipids at the expense of reduced glutathione (GSH), which is oxidized in the process. GSH is a tripeptide produced from cysteine, glycine, and glutamate in the cytoplasm. Cysteine enters the cell through the cystine-glutamate exchanger known as the antiporter system x_c_^-^. Inhibition of cysteine import and depletion of the cellular GSH pool by system x_c_^-^ inhibitors (e.g., erastin) have been shown to impair GPX4 function, leading to ferroptosis [4]. Another classic inducer of ferroptosis is the RAS-selective lethal 3 (RSL3), which promotes ferroptosis through the direct inhibition of GPX4 activity [7].

Recent studies suggested a direct link between ferroptosis and coronary heart diseases, particularly myocardial infarction and ischemia-reperfusion, in animal models and patients [8,9,10,11,12]. However, the mechanisms of ferroptosis signaling in the heart remain unknown, which represents a challenge for the development of new therapeutic strategies for coronary heart diseases. Inhibition of ferroptosis by the synthetic antioxidant ferrostatin-1 (Fer-1) demonstrated strong protective effects against cell death in cultured cardiomyocytes and isolated perfused hearts [8,9,10,12,13]. Fer-1 inhibits lipid peroxidation more effectively than other phenolic antioxidants since it scavenges alkoxyl radicals without being consumed in the process and then is regenerated by ferrous iron, reducing back the Fer-1 radical [14]. Furthermore, oxidized phosphatidylethanolamine, the major ferroptotic signaling phospholipid, was visualized in cultured H9c2 cardiomyocytes [15]. These studies suggest that the target components involved in ferroptosis are localized in cardiomyocytes. Ferroptosis induces significant alterations in cellular metabolism, including metabolites that are required for GSH synthesis and are involved in the synthesis of tricarboxylic acid (TCA) cycle intermediates in mitochondria [13,16]. In addition, lipid metabolism plays a regulatory role in cell death [17]. Monounsaturated FA, such as oleic and palmitoleic acids have been shown to potently suppress the toxic environment caused by lipid peroxidation in the plasma membrane, hence inhibiting ferroptosis [17]. On the other hand, saturated membrane lipids were less sensitive to oxidative stress in cancer cells, suggesting that high levels of the different saturated membrane phospholipids can protect against oxidative damage [18]. Altogether, these studies highlight the importance of evaluating the levels of different cell metabolites that can be affected during ferroptosis.

Mitochondria play a central role in the pathogenesis of human diseases [19]. They contain the main components and enzymes of the ferroptotic machinery, such as lipoxygenase, GPX4, GSH, glutamate, and iron. Mitochondria do not synthesize GSH and are therefore dependent on the synthesis of cytosolic GSH [20,21]. However, glutamate, a product of glutaminolysis in mitochondria, maintains the intracellular cysteine levels through the activation of the antiporter system x_c_^−^ and thus, the cellular GSH pool. Glutaminolysis is a metabolic pathway that involves the initial deamination of glutamine by glutaminase C (GAC), a K-type mitochondrial glutaminase (GLS), which is highly expressed in the heart and pancreas [22,23]. Furthermore, besides GSH precursors, the role of other cellular and mitochondrial metabolites that could be altered in response to the ferroptotic stimuli remains vaguely understood.

In this study, we elucidated the effects of ferroptosis on the metabolome in H9c2 cardioblasts by gas chromatography-mass spectrometry (GC-MS). Besides Fer-1, our study implemented the use of XJB-5-131 (XJB), a synthetic radical and electron scavenger which directly targets mitochondria, and its analog TSM-1005-44 (TSM), a newly developed anti-ferroptotic compound. Our results demonstrated that the metabolome was affected by RSL3-induced ferroptosis. Particularly, all three amino acids involved in GSH synthesis were drastically reduced in the RSL3-treated group. Ferroptosis impaired mitochondrial metabolism is evidenced by diminished succinate levels and high isocitrate and 2-oxoglutarate levels, which led to increased levels of saturated FA. In addition, genetic inhibition of glutaminolysis provided initial beneficial effects and protected the cells against RSL3-induced ferroptosis. This study demonstrates that cellular and mitochondrial metabolism is highly sensitive to ferroptotic stimuli.

## 2. Materials and Methods

### 2.1. Physico-Chemical Parameters of TSM-1005-44 (TSM)

The new anti-ferroptotic compound, TSM (4-[[(3*E*,5*S*)-5-[[(1,1-dimethylethoxy)carbonyl]amino]-2,2-di(4-fluorobenzyl)-5-(4-fluorophenyl)-1-oxo-3-penten-1-yl]amino]-2,2,6,6-tetramethyl-1-piperidinyloxy) is a ca. 10-fold more potent analog of JP4-039. Characteristic spectroscopic data are: ${\left[\alpha \right]}_{D}^{25}$ −23.5 (*c* 0.1, CHCl_3_); LCMS ESI^+^ *m/z* hydroxylamine [M + H]^+^ 680.3, Rt 6.05 min; nitroxide [M + H]^+^ 679.3 and [M + Na]^+^ 701.3, Rt 7.13 min; HRMS (ESI) *m/z* calculated for C_39_H_48_N_3_O_4_F_3_ [M + H]^+^ 679.3591, found 679.3567. Preliminary studies demonstrated that TSM, similar to XJB, prevents ferroptotic cell death with an EC_50_ of 300–600 nM, depending on the cell line and the ferroptosis inducer (RSL3 or erastin). Significantly, TSM has several physico-chemical advantages over XJB, increasing its drug-likeness score and rendering this analog a superior candidate for further biological testing and clinical candidate development: (a) a simplified structure (f. ex. −4 stereocenters vs. XJB), which allows for more expedient chemical synthesis; (b) a lower molecular weight (−300 units vs. XJB); (c) a lower TPSA (−120 units vs. XJB); (d) a lower cLogP and (e) a higher aqueous solubility.

### 2.2. Cell Culture

H9c2 cardioblasts were cultured according to the manufacturer’s recommendations (ATCC, Manassas, VA, USA) with minor modifications [12]. Briefly, the cells were incubated in DMEM based modified media containing 4 mM L-glutamine, 4.5 g/L glucose, 1 mM sodium pyruvate, and 1.5 g/L sodium bicarbonate, pH 7.4, and supplemented with 10% fetal bovine serum and 1% antibiotic solution (HyClone, Logan, UT, USA) in a CO_2_ incubator containing 95% air and 5% CO_2_ at 37 °C. Cells with 80–90% confluence from passages 3–10 were used for experiments. All chemicals were purchased from Sigma-Aldrich (St. Louis, MO). H9c2 cardioblasts demonstrate similar to the primary cardiomyocytes hypertrophic response [24] as well as mitochondrial metabolism and morphology [25].

### 2.3. Cell Treatments

Cultured H9c2 cells reaching 80–90% confluence from passages 3–10 were used for experiments. Ferroptosis was induced by incubating the cells with 0.5 µM RSL3 for 3 h. In addition, the cells were exposed to RSL3 in the presence of 1 µM Fer-1, 0.2 µM XJB, and 0.6 µM TSM. Thus, experiments were performed in the following 5 groups: (i) control (*n* = 5), (ii) RSL3 (*n* = 6), (iii) RSL3 + Fer-1 (*n* = 6), (iv) RSL3 + XJB (*n* = 6), and (v) RSL3 + TSM (*n* = 5). Fer-1, XJB, or TSM were added to the culture medium simultaneously with RSL3. At the end of the incubation, the cells were harvested and processed for analysis by GC-MS.

### 2.4. Metabolites Extraction

Metabolites were extracted as previously described [26] using 1 mL of cold MeOH/H_2_O (85:15). Samples were sonicated for 15 s (3x) on ice and centrifuged at 1400 rpm × 10 min at 4 °C (Rotor model: Eppendorf, FA-45-30-11). For protein quantification, the pellets were dried for 15 min in a rotary vacuum evaporator, then resuspended in 60 μL of denaturation buffer and sonicated for 1 min. All samples were centrifuged for 10 min at the maximum speed and the supernatant was used for protein quantification using the Bradford method. Supernatants were collected and evaporated to dryness using a SpeedVac (Savant AS160, Farmingdale, NY, USA). The metabolite samples were first derivatized through methoxyamination by adding 30 μL of 20 mg/mL solution of methoxyamine hydrochloride in pyridine (Sigma-Aldrich, St. Louis, MO, USA) and incubated at 37 °C for 2 h. Afterward, trimethylsilylation was performed by adding 30 μL of N-tert-butyldimethylsilyl-N-methyltrifluoroacetamide (MTBSTFA + 1% TBDMSCl; Sigma-Aldrich, St. Louis, MO, USA) and incubated for 1 h at 65 °C. The reaction mixture was centrifuged at 1300 rpm × 10 min at RT. Supernatants were transferred to analytical vials. Then, 20 μL per sample was added to glass vials with inserts followed by the addition of 1 mM 2-fluobiphenyl (Sigma-Aldrich, St. Louis, MO, USA) as an internal standard. Samples were processed by GC-MS-QP2010 (Shimadzu Scientific Instruments, Inc., Columbia, MD) using analytical conditions as previously described [27].

### 2.5. Data Processing and Bioinformatics

Raw chromatography data were obtained and processed in GC-MS Labsolution Postrun Analysis software (Shimadzu Scientific Instruments, Inc., Columbia, MD) for identification of metabolites from their electron impact mass spectra by comparison to the database (NIST14/2014/EPA/NIH) [26]. Peak integration for all metabolites and multiple searches in the mass spectral library database resulted in a final database of 52 metabolic features chosen for this analysis. To assess analytical accuracy and precision, an external quality evaluation was performed using 2-fluorobiphenyl spiked into derivatization blank samples before running on the GC-MS. The quantitative analysis corresponding to the metabolite concentrations in each sample was calculated based on the internal standard in mM. A table in the format of comma-delimited (* csv) was created and uploaded to MetaboAnalyst.ca [28]. The latter was used for data processing and statistical analysis, including data normalization. The samples were normalized to protein content, previously calculated in each sample, and range scaled. The heatmap plot was generated in the MetaboAnalyst (5.0) platform [28], where different conditions and bins were clustered and organized in dendrograms, using hierarchical clustering by the Ward-linkage method and Euclidean distances. To quantify the differences between average groups, a multivariate Euclidean distance (with autoscale) was performed between the control and other conditions. The resulting distances were scaled and translated into a color code for a graph connecting the different adjacent conditions. Multivariate analysis between all treatment groups and control groups was performed by the default parameters of MetaboAnalyst.ca of one-way ANOVA with Fisher’s LSD post hoc test. A *p* < 0.05 was considered statistically significant and values were subjected to the Bonferroni correction.

### 2.6. SiRNA Transfection

To downregulate *GLS1*, which encodes for the GAC protein, a mitochondria-localized K-type GLS responsible for converting glutamine to glutamate, H9c2 cardioblasts were transfected using Lipofectamine RNAiMAX (Invitrogen, Thermo Fisher Scientific, Waltham, MA, USA) and FlexiTube small interfering RNA (siRNA, Qiagen, Germantown, MD, USA) according to the manufacturer’s recommendations. Briefly, the cells were seeded with Opti-MEM ™ Reduced Serum Medium, GlutaMAX ™ (Thermo Scientific, Thermo Fisher Scientific, Waltham, MA, USA) and supplemented with 5% fetal bovine serum and 1% antibiotic solution. Once cell density reached the 40–60% confluency range after 24 h, Lipofectamine RNAiMAX (Invitrogen, Thermo Fisher Scientific, Waltham, MA, USA) and FlexiTube siRNA mixtures were added. All experiments were conducted 48 h post-transfection. The siRNA sequences were UUCUCCGAACGUGUCACG (non-coding, NC) and AACCACATAATCCGATGGTAA (*GAC*).

### 2.7. Cell Death Assay

The activity of lactate dehydrogenase (LDH) in the incubation medium (cell culture studies) was determined as a marker for cell death. The LDH activity was measured by spectrophotometry at 25 °C as described previously [29]. Data were normalized to the control, and results are represented as a fold of increase.

### 2.8. SDS-PAGE and Western Blotting

To analyze protein levels, equal amounts of protein were resolved by SDS-PAGE and transferred to nitrocellulose membranes (GE Healthcare Bio-Sciences, Pittsburgh, PA, USA). The membranes were immunoblotted with antibodies against *GAC* (Abcam, ab-202027), GAPDH (Santa Cruz, sc-32233), ATP5a (Abcam, ab-14748, Cambridge, MA, USA), and GPX4 (Abcam, ab125066), followed by incubation with IRDye ^®^ (LI-COR Biosciences, Lincoln, NE, USA) secondary antibodies. Bands were visualized using an ODYSSEY ^®^ CLx (LI-COR Biosciences, Lincoln, NE, USA) infrared scanner. The resulting images were analyzed with Image Studio Lite Software version 5.2, and results are represented as a percentage of NC.

### 2.9. Statistical Analysis

Data corresponding to each group were analyzed using two- and one-way ANOVA with a normality test (Shapiro-Wilk) and a multiple comparison procedure with Bonferroni correction, in addition to Student’s *t*-test for independent comparison in between samples. Results are presented as mean ± SE. Where *p <* 0.05 was considered statistically significant.

## 3. Results

### 3.1. The Effects of RSL3 on Cell Death

First, we established optimal concentrations of XJB and TSM that could exert maximum anti-ferroptotic effects in H9c2 cells. The cells were exposed to a ferroptotic stimulus by treatment with RSL3 (0.5 µM) for 3 h in the absence and presence of XJB or TSM in a concentration range from 0.1 to 1 µM. Results showed that XJB reached a maximum protective effect at 0.1 µM (Figure 1A), whereas TSM demonstrated the highest protection starting at 0.6 µM (Figure 1B). Next, based on the LDH activity in the cell culture medium to assess cell death, we determined an optimal exposure time in which H9c2 cardiomyocytes developed ferroptosis once exposed to RSL3. The cells were incubated with 0.5 µM RSL3 for 1, 2, and 3 h in the absence and presence of Fer-1, XJB, or TSM (Figure 2). Incubation of the cells with RSL3 for 1 and 2 h had no significant effects on cell survival, as evidenced by no significant differences in LDH activity between control and RSL3-treated groups. However, RSL3 induced approximately a 2.3-fold increase (*p* < 0.001) in LDH activity after 3 h of incubation compared to the control. Fer-1, XJB, and TSM protected the cells against RSL3-induced cell death and prevented LDH release to the culture medium. The antioxidants alone (no RSL3-treatment) had no detrimental effects on the cells (Appendix A).

### 3.2. Identification of Metabolites

To identify and quantify cellular metabolites, cell lysates obtained from H9c2 cardioblasts after incubation with 0.5 µM RSL3 for 3 h in the absence and presence of Fer-1, XJB, and TSM were processed through GC-MS. A total of 52 metabolites were identified and classified in the following groups: amino acids (44%), TCA metabolites (11%), saturated FA (9%), unsaturated FA (6%), pyrroline carboxylic acids (6%), and 2% of pyrimidines, secondary, and sugar alcohols, short-chain acids, and cholesterol, among others (Figure 3A). The characterization of each identified metabolite, including chemical names, fragment ions, retention time, and classification, is detailed in Appendix A, whereas information regarding the statistics that include the p.values and fold of change can be found in Appendix A. The heatmap shows the results obtained from univariate analysis of metabolite concentration samples, considering the intensity of the color, as discrete variables (−2 to 2), by assorting them in bins (Figure 3B). The conditions (in the abscise axis) and the bins/metabolites (in the ordinate axis) have been arranged according to the Ward-clustering method. The fold change is color-coded according to the bar legend: a red/green value in the heatmap indicates an increase/decrease in average concentrations of the metabolites. The heatmap shows data clusters based on group similarities in which two superior clusters were identified. The first group contains only metabolites in the presence of RSL3, and the second group (control, TSM, RSL3 + XJB, Fer-1, RSL3 + TSM, RSL3 + Fer-1, and XJB) was clustered based on their data value proximity. Data shows the similarities of metabolites between the control and TSM; together with RSL3 + XJB, and Fer-1 with RSL3 + TSM. The latter altogether compares to a separate cluster containing RSL3 + Fer-1 and XJB. These results show how the RSL3-challenged group separates from the rest, indicating significant up- or downregulation of metabolites during ferroptosis. Interestingly, the control group shares a similar metabolite concentration with the TSM-treated group, while RSL3 + Fer-1 and XJB groups differentiate from RSL3 with the most distance, indicating opposing metabolite regulation compared to RSL3 alone. A one-way ANOVA followed by a post hoc test analysis revealed a total of 22 metabolites that had statistically significant differences in their concentrations between the treatment groups and the control. After performing Bonferroni correction, 12 metabolites were identified with a new *p*-value being *p* = 0.0022: stearic acid, glycolic acid, succinate, proline, palmitic acid, azelaic acid, beta-alanine, aspartic acid, glycine, alanine, (2*R*)-pyrrolidine 1,2-dicarboxilic acid [shown as (2*R*)-pyrrolidine in Figure 3B and Appendix A], and lysine (Appendix A). Particularly, in comparison with the control group, RSL3-treated metabolites’ concentration was increased in azelaic, glycolic, palmitic, and stearic acid, and decreased the levels of other metabolites. No changes were found in groups treated with RSL3 in combination with Fer-1 and TSM compared to the control. RSL3, in combination with TSM, decreased the concentrations of azelaic, glycolic, palmitic, and stearic acid slightly; and the same effect was observed in Fer-1 alone. XJB reduced the levels of all metabolites except azelaic, glycolic, palmitic, and stearic acid. In contrast, RSL3 supplemented with XJB reduced proline, (2*R*)-pyrrolidine 1,2-dicarboxilic acid, alanine, beta-alanine, glycine, and aspartate.

### 3.3. Effects of Ferroptosis on Cell Metabolites

#### 3.3.1. GSH Precursor Amino Acids

Taking into consideration the important role that GSH plays in the development of ferroptosis, we sought to evaluate the precursor metabolites of GSH, which are glycine, cysteine, glutamate, and glutamine, and its degradation byproducts (5-oxoproline) (Figure 4). RSL3 decreased glycine, cysteine, and glutamine levels by 60% (*p* < 0.05), 64% (*p* < 0.001) and 76% (*p* < 0.05), respectively, compared to the control. No effects were observed in glutamate and 5-oxoproline levels in the presence of RSL3. Treatment with Fer-1 increased glycine, cysteine, and glutamine by 93% (*p* < 0.01), 94% (*p* < 0.05) and 141% (*p* < 0.001), respectively, in the presence of RSL3, with no changes in glutamate and 5-oxoproline. Additionally, XJB increased the levels of glutamine by 81% (*p* < 0.05). Similarly, to Fer-1, TSM significantly increased the levels of glycine by 96% (*p* < 0.01) and glutamine by 197% (*p* < 0.001) with no changes in other associated metabolites.

#### 3.3.2. Fatty Acids

Next, we evaluated the effects of ferroptosis in saturated and unsaturated FA. Analysis of saturated FA revealed that RSL3 increased the levels of capric, myristic, palmitic, and stearic acids by 184% (*p* < 0.001), 83% (*p* < 0.05), 126% (*p* < 0.001) and 153% (*p* < 0.001), respectively, compared to the control group (Figure 5A). However, treatment of the cells with Fer-1, XJB, or TSM significantly attenuated the effects of RSL3 on the saturated FA levels that were similar to the control group. Unsaturated FA exhibited a certain trend of increasing in RSL3-challenged cells, although the effect was not statistically significant (Figure 5B).

#### 3.3.3. TCA Metabolites

RSL3 induced significant changes in the levels of TCA intermediates, including citrate, isocitrate, 2-oxoglutarate, succinate, fumarate, and malate (Figure 6A). RSL3 increased isocitrate and 2-oxoglutarate levels by 130% (*p* < 0.01) and 73% (*p* < 0.05), respectively, compared to the control, whereas the level of succinate was decreased by 55% (*p* < 0.001). RSL3 had no significant effects on other TCA intermediates. Treatment with Fer-1, XJB, and TSM attenuated RSL3-induced changes in TCA metabolites, particularly, isocitrate, 2-oxoglutarate, and succinate levels. Treatment with Fer-1 and TSM decreased 2-oxoglutarate levels by 58% (*p* < 0.05) and 76% (*p* < 0.05), respectively, compared to the RSL3 group. Likewise, RSL3-induced increase in isocitrate levels was completely prevented in the presence of TSM. All three antioxidants significantly increased the levels of succinate in RSL3-challenged cells by 140% (*p* < 0.01), 130% (*p* < 0.01), and 113% (*p* < 0.05) for Fer-1, XJB, and TSM, respectively. Fumarate levels were decreased by 52% (*p* < 0.05) and 57% (*p* < 0.05), respectively, in Fer-1- and TSM-treated cells compared to the RSL3 group (Figure 6A). Thus, we observed significant changes in the levels of TCA metabolites induced by RSL3 that were ameliorated in the presence of Fer-1, XJB, and TSM.

#### 3.3.4. Amino Acids

Thereafter, we evaluated the effect of ferroptosis on the level of amino acids (isoleucine, alanine, leucine, and methionine) that are related to energy metabolism. We found that RSL3 did not change the levels of these amino acids compared to the control group (Figure 6B). However, treatment with Fer-1 significantly increased the levels of alanine by 36% (*p* < 0.05) and methionine by 53% (*p* < 0.05), compared to RSL3, with no changes in isoleucine and leucine. XJB did not affect the levels of any of the metabolites, whereas treatment with TSM significantly increased the levels of alanine by 43% (*p* < 0.05) in RSL3-challenged cells, with no changes in isoleucine, leucine, and methionine.

### 3.4. Glutaminolysis Is Involved in Ferroptotic Signaling in Mitochondria

In this section, we examined the effect of glutaminolysis in mitochondria isolated from H9c2 cardiomyocytes exposed to ferroptosis, and the activity of LDH in the medium was measured to estimate cell death at 1 and 3 h. First, the expression of the mitochondrial glutaminase GAC in the cells was silenced with a *GLS1* siRNA. Control (NC, non-coding siRNA-treated) and *GLS1* knockdown cells were incubated with 0.5 µM RSL3 to induce ferroptosis, in the absence or presence of 1 µM Fer-1. Results showed a 51% (*p* < 0.05) reduction of GAC protein levels in the cells treated with *GLS1* siRNA compared to NC cells (Appendix A). Analysis of LDH release in NC showed no changes in LDH activity compared to the control at 1 h. However, in *GLS1* knockdown cells, LDH activity was reduced in the RSL3 group, compared to NC. (Figure 7A). In contrast, the activity of LDH from RSL3-challenged cells was significantly increased compared to the control in both NC and *GLS1* knockdown cells after 3 h of incubation (Figure 7B). Likewise, GSH levels were higher in *GLS1* knockdown cells in response to RSL3 at 1 h; however, the effects were lost at 3 h of incubation (Figure 7C,D). These data suggest that glutaminolysis is involved only at the early stage (1 h) of ferroptotic signaling.

## 4. Discussion

In this study, we demonstrated that ferroptosis induced by RSL3 results in dramatic changes in the metabolic profile of H9c2 cardiomyocytes. LDH activity in RSL3-challenged cells was significantly increased after a 3 h incubation period, indicating high cell death in response to the ferroptotic stimuli. However, RSL3 was not able to induce cell death in the presence of several structurally diverse antioxidants (Fer-1, XJB, and TSM). We have previously provided direct evidence of the accumulation of oxidized phospholipids in H9c2 cardiomyocytes during RSL3-induced ferroptosis. In these studies, a direct mapping of oxidized phospholipids was performed by the gas cluster ion beam secondary ion mass spectrometry (GCIB-SIMS) imaging technique, which allowed us to identify oxidized phospholipids with a 1.2 μm spatial resolution at the single-cell level [15]. In the present study, RSL3 diminished GPX4 protein expression, an effect that was prevented by Fer-1 and TSM. For the first time, we revealed the anti-ferroptotic effectiveness of TSM, a newly developed cellular antioxidant and ROS scavenger, against RSL3-induced ferroptosis in cardiomyocytes. The effects of the mitochondrial-targeted ROS scavenger, XJB, observed in this study are consistent with our previous findings where XJB exhibited anti-ferroptotic effects in HT-1080 fibrosarcoma cells exposed to RSL3 or erastin [30] and in iron- or BSO-treated murine I154F fibroblasts [31]. Additionally, we have previously shown cardioprotective [32,33] and antiaging [34] effects of XJB in rats. The similarities between the biological effects of the three chemically and structurally diverse antioxidants, Fer-1, XJB, and TSM, suggest that the ferroptotic signaling pathway in H9c2 cardiomyocytes is most likely triggered in the cytosol, but that protection of mitochondria is sufficient to stop it.

Intriguingly, RSL3 triggered dramatic decreases in the level of GSH precursor amino acids (glycine, cysteine, and glutamine) which was completely prevented in the cells treated with Fer-1, XJB, or TSM. These data indicate that downregulation of GSH biosynthesis due to the deficiency of precursor amino acids may serve as a driving factor in RSL3-induced ferroptosis. RSL3 significantly augmented the level of saturated FA, but had no effects on unsaturated FA, suggesting a high sensitivity of saturated FA metabolism to ferroptotic stimuli. High levels of saturated FA can be explained by alterations in the metabolism of saturated FA and their conversion to unsaturated FA. Recent studies have demonstrated that monounsaturated FA induces a ferroptosis-resistant state in the cells that was dependent on the long-chain acyl-CoA synthetase family member [35]. The study found that monounsaturated FA were not able to increase the expression of GPX4, which inhibits ferroptosis, but they attenuated the accumulation of oxidized lipids in plasma. Moreover, caspase-dependent lipotoxicity of saturated FA was suppressed by exogenous monounsaturated FA [35]. Results of our study suggest that increased levels of saturated FA (Figure 5A) could result from inhibition of SCD1, an enzyme responsible for the unsaturation of stearic acid to oleic acid. In favor of this, SCD1 inhibition has been shown to participate in the induction of ferroptosis signaling by decreasing both CoQ_10_ and unsaturated FA chains [36]. The connection between RSL3 and the accumulation of saturated FA is poorly understood. Further studies are required to elucidate whether the accumulation of saturated FA can stimulate ferroptotic signaling pathways.

RSL3 affected TCA cycle intermediates, as evidenced by increased isocitrate and 2-oxoglutarate levels and decreased the level of succinate. These data and the findings resulting from *GLS1* knockdown experiments (Figure 7) demonstrate the importance of the glutaminolysis-TCA axis in ferroptosis in H9c2 cardiomyocytes. The TCA cycle intermediates (2-oxoglutarate, succinate, malate, and fumarate) were previously shown to increase lipid ROS accumulation and promote ferroptosis induced by cysteine-deprivation or erastin in both mouse embryonic fibroblasts (MEFs) and fibrosarcoma HT-1080 cells [37]. This can be explained, in part, by enhanced mitochondrial ROS production by the TCA cycle enzymes (e.g., oxoglutarate dehydrogenase) and ETC complexes that, in turn, could further amplify ferroptotic signaling. These studies demonstrated that inhibitors of ETC complexes were protective against cysteine-deprivation and erastin-induced ferroptosis, and significantly reduced cell death and lipid peroxidation [37]. In contrast, RSL3-induced ferroptosis in H9c2 cardiomyocytes was further accelerated by inhibition of ETC complexes and OXPHOS [12]. The discrepancy between previous studies and ours can be explained by the differences in cell type, ferroptosis inducers, and exposure time. Particularly, activation of ferroptotic signaling in response to ferroptosis inducers (RSL3, erastin, and cysteine-deprivation) might be mediated through different mechanistic pathways in mitochondria.

Mitochondrial glutaminolysis acts as a major source of anaplerosis and fuels the TCA cycle with 2-oxoglutarate through the oxoglutarate dehydrogenase [38]. Glutamine, in addition to GSH, participates in many metabolic pathways, including the biosynthesis of essential metabolites in the cell. Due to a crucial role in energy metabolism required for cell growth and differentiation, glutamine is consumed at high levels in cancer cells [38,39] that are, accordingly, very sensitive to ferroptosis induced by glutamine deficiency. GLS catabolizes glutamine to glutamate, whereas glutamic-oxaloacetic transaminase is responsible for the conversion of glutamate into 2-oxoglutarate. Genetic silencing and pharmacological inhibition of glutaminolysis through GLS2 and glutamic-oxaloacetic transaminase have been shown to inhibit erastin-induced ferroptosis [8,40]. Our findings on the effect of *GLS1* silencing on RSL3-induced cell death are partially consistent with these studies. We observed early anti-ferroptotic effects of *GLS1* inhibition that disappeared at the late stages of ferroptosis in H9c2 cardiomyocytes. Indeed, we have previously reported an early response (30 min) of mitochondria to ferroptotic stimuli induced by RSL3 in H9c2 cardiomyocytes [12]. In addition, the levels of ferroptotic oxidized phospholipids were significantly higher in mitochondria compared to the cytosol. These findings suggest a critical role of mitochondria in the propagation of ferroptotic signaling that can be lost with the progression of the death stimuli. Mitochondrial ferroptotic signaling can stimulate the ferroptosis machinery in the cytoplasm or develop independently.

In addition to glutaminolysis, other mechanisms might be involved in the regulation of ferroptosis. The transsulfuration pathway, which transfers sulfur from homocysteine to cysteine, is another mechanism that regulates the level of GSH. Low cysteine levels found in our study can be associated with decreased formation of Fe-S clusters, which consequently increases the free iron pool inside the mitochondria. It has been demonstrated that low cysteine levels lead to decreased Fe/S cluster biosynthesis, which induces an iron starvation response that eventually promotes more iron import thus, enhancing ferroptosis [41].

In conjunction with the inactivation of GPX4, free iron can further activate phospholipid oxidation in the plasma/mitochondrial membrane and inevitably lead to ferroptosis [41]. Interestingly, TfR1 receptors, which transport intracellular iron into the cells, were augmented under the same conditions [42], contributing to ferroptosis signaling [43]. Nevertheless, the mechanisms that promote the increase of TfR1 receptors during ferroptosis remain obscure. Mitochondrial proteins, such as iron regulatory proteins, have been thought to be the mediator of this process by interacting with iron response elements to promote the synthesis of TfR1 receptors [44,45].

Increased levels of isocitrate in cardiomyocytes in response to RSL3 can be due to the downregulation of isocitrate dehydrogenase (IDH), an NADP^+^-dependent enzyme, which requires GSH as a substrate to maintain its redox state [46]. Increased depletion of GSH due to its reduced biosynthesis and increased utilization in response to ferroptotic stimuli presumably inhibits IDH activity and reduces isocitrate levels. In favor of this, IDH knockdown in HT-1080 fibrosarcoma and Hepa1-6 hepatoma cells promoted erastin-induced GPX4 inhibition [47,48]. In contrast, succinate levels were found to decrease during RSL3-induced ferroptosis in H9c2 cells. We suggest that succinyl-CoA, a precursor molecule for succinate, could participate in the inhibition of IDH via a post-translational modification named succinylation, thus contributing to the increase of isocitrate levels. Furthermore, succinyl-CoA can be consumed for the formation of heme as an iron reservoir mechanism through which succinate is being consumed [49,50]. Low succinate levels in RSL3-challenged cells can compromise ETC complex II activity and thereby, increase the FADH_2_ to FAD^+^ ratio. As a result, the activity of the short-chain acyl-CoA dehydrogenase, the rate-limiting enzyme for oxidation of saturated FA that requires FAD^+^ as a cofactor, can be compromised, leading to the increased levels of saturated FA (Figure 5).

## 5. Conclusions

In conclusion, the analysis of the cellular metabolome revealed that a variety of different metabolic pathways, including the metabolism of amino acids and FA, mitochondrial bioenergetics, and glutaminolysis, are involved in ferroptotic signaling in H9c2 cardiomyocytes. We showed for the first time that the anti-ferroptotic effects of a recently developed antioxidant and cellular ROS scavenger, TSM, in cardiomyocytes were similar to those of the structurally different Fer-1 and XJB antioxidants. Despite the great advancements that have been made in determining the role of ferroptosis in cardiac diseases, this cell death pathway is only vaguely understood. For this reason, more studies need to be conducted to further elucidate the relationship between the pathways discussed in this study and other possible undiscovered mechanisms underlying ferroptotic signaling in cardiac cells.

### Limitations

The scope of our study was to conduct an untargeted metabolomics analysis and identify the effects of ferroptosis on cardiac cells’ metabolism; however, our study contains certain limitations. First, a metabolic flux analysis was not conducted to determine the specific pathways involved in ferroptosis. Second, we did not test the combinatory effects of the antioxidants used in this study in RSL3-challenged cells. Third, we used H9c2 cardioblasts instead of primary cardiomyocytes. Finally, fourth, we did not assess the contribution of the changes in the antioxidant capacity to factors not related to cell death, such as hypertrophy, electrical signaling, and/or inflammation.

## Figures and Tables

**Figure 1 antioxidants-11-00278-f001:**
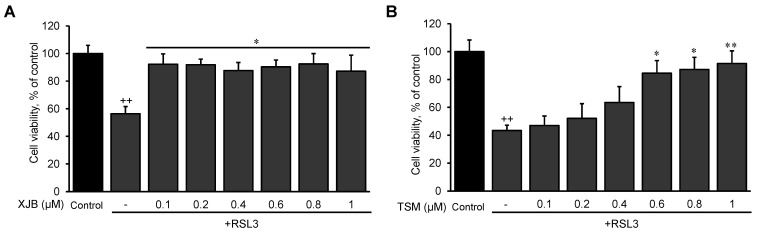
The effects of XJB and TSM on ferroptosis in H9c2 cells. Different concentrations of XJB (**A**) and TSM (**B**) in the presence or absence of 0.5 µM RSL3. Cell death was calculated from the measurements by the Alamar Blue cell viability assay, and data are presented as a percentage of the control. GPX4 protein levels were calculated based on densitometry analysis using LI-COR Image Studio Lite and normalized to β-actin expression for each band. Data are presented as a percentage of the control group. ^++^
*p* < 0.01 vs. control; * *p* < 0.05 and ** *p* < 0.01 vs. RSL3. *n* = 3 per group.

**Figure 2 antioxidants-11-00278-f002:**
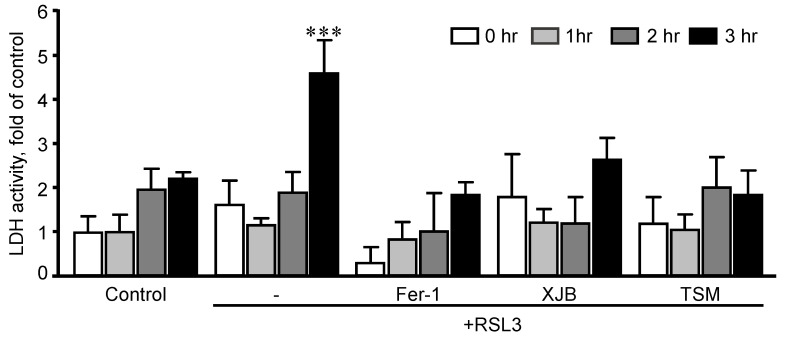
Time-dependence analysis of RSL3-induced cell death in H9c2 cells. LDH activity as a marker of cell death was measured in the culture medium containing H9c2 cells that were exposed to 0.5 µM RSL3 for 1, 2, and 3 h, in the presence and absence of 1 µM Fer-1, 0.2 µM XJB, or 0.6 µM TSM. Data were normalized to the control for each time-point and presented as a fold increase. *** *p* < 0.001 vs. control. *n* = 3 per group.

**Figure 3 antioxidants-11-00278-f003:**
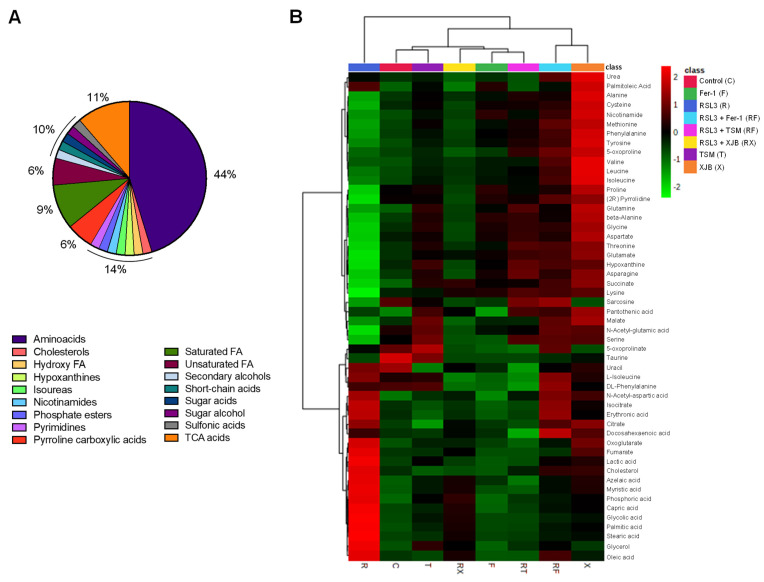
The metabolome of H9c2 cells exposed to ferroptosis. The constituent ratio of the 52 metabolites is identified by class (**A**). Heatmap for the different conditions as compared to the control and RSL3 displaying average metabolites concentration (mM) for each group (**B**). The conditions (in the abscise axis) and the bins/metabolites (in the ordinate axis) have been sorted according to cluster analysis. The fold change is color-coded according to the bar legend. C, control; R, RSL3; F, Fer-1; X, XJB; T, TSM; RF, RSL3 + Fer-1; RX, RSL3 + XJB; RT, RSL3 + TSM. Green represents lower concentration; red represents higher concentrations. *n* = 4–6 per group.

**Figure 4 antioxidants-11-00278-f004:**
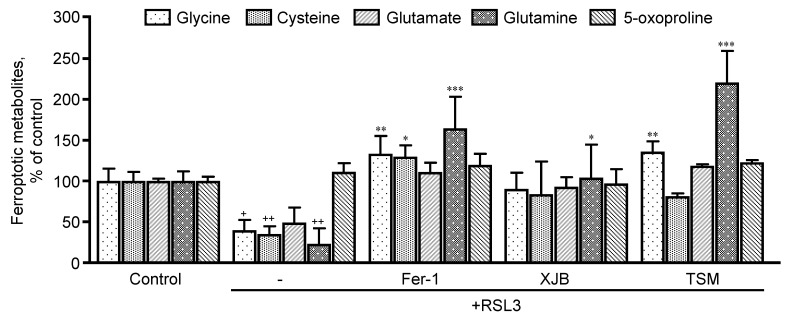
Precursor metabolites of GSH metabolism. Glycine, cysteine, glutamate, and glutamine represent GSH precursors, while 5-oxoproline is a marker of GSH degradation within the cell. Data are presented as a percentage of the control group. ^+^
*p* < 0.05, ^++^
*p* < 0.01 vs. control; * *p* < 0.05, ** *p* < 0.01 and *** *p* < 0.001 vs. RSL3. *n* = 4–6 per group.

**Figure 5 antioxidants-11-00278-f005:**
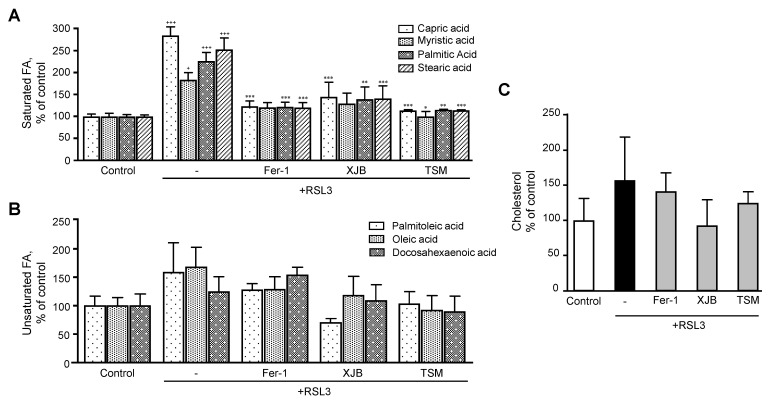
Fatty acids and cholesterol levels after ferroptosis. (**A**) Saturated FA. (**B**) Unsaturated FA. (**C**) Cholesterol. Data are presented as a percentage of the control group. ^+^ *p* < 0.05 and ^+++^ *p* < 0.001 vs. control; * *p* < 0.05, ** *p* < 0.01 and *** *p* < 0.001 vs. RSL3. *n* = 4–6 per group.

**Figure 6 antioxidants-11-00278-f006:**
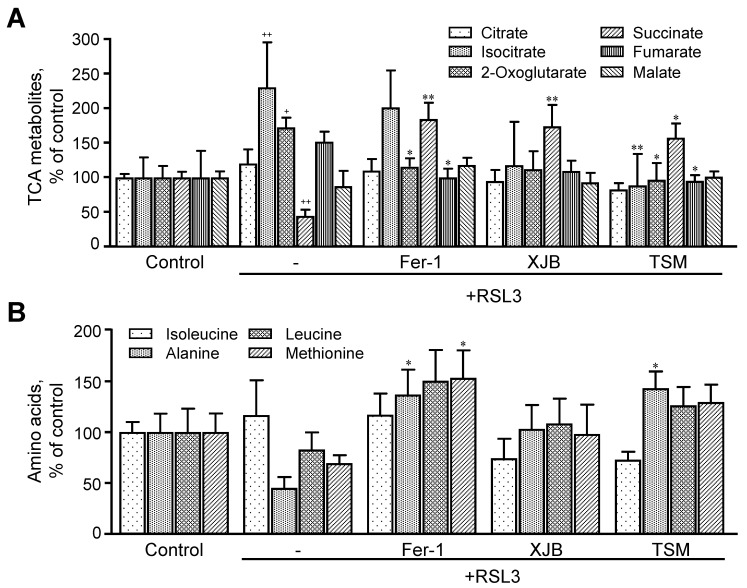
TCA metabolites and amino acids levels after ferroptosis. The TCA metabolites (**A**) identified were citrate, isocitrate, 2-oxoglutarate, succinate, fumarate, and malate. The amino acids represented are isoleucine, alanine, leucine, and methionine (**B**). Data are presented as a percentage of the control group. ^+^
*p* < 0.05, ^++^
*p* < 0.01 vs. control; * *p* < 0.05 and ** *p* < 0.01 vs. RSL3. *n* = 4–6 per group.

**Figure 7 antioxidants-11-00278-f007:**
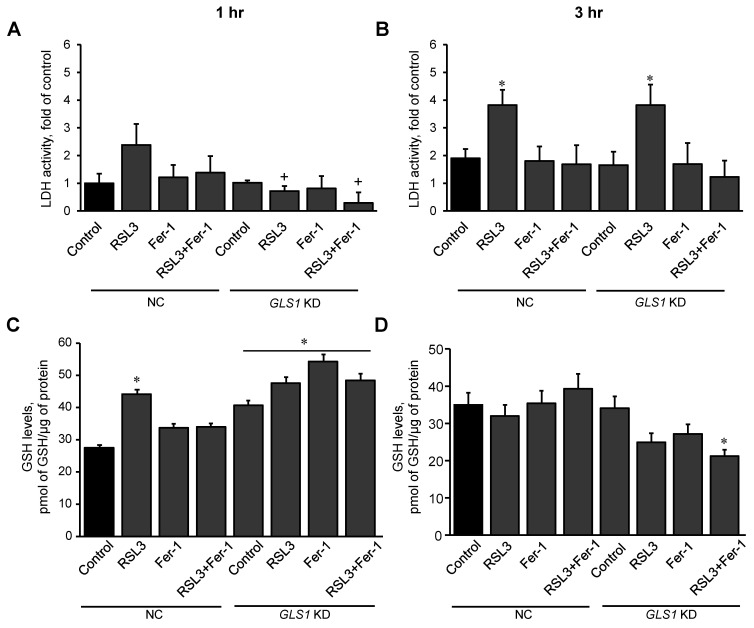
RSL3-induced cell death and GSH levels in *GLS1* KD H9c2 cells after ferroptosis. H9c2 cells from non-coding (NC) and GLS KD groups were incubated in the presence and absence of 0.5 µM RSL3, 1 µM Fer-1, or both for 1 (**A**) and 3 h (**B**). LDH activity, as a marker of cell death, was measured in the culture medium containing H9c2 cells. Results were normalized to the control. GSH was quantified in mitochondria isolated from the cells after 1 hr (**C**) and 3 hr (**D**) of incubation. Data were normalized to a standard curve of GSH which ranged 1-10 µM. * *p* < 0.05 vs. control (NC), ^+^
*p* < 0.05 vs. RSL3 (NC). *n* = 3 per group.

## Data Availability

Data is contained within the article and Appendix A.

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
