# Peer review of "Effects of Ferroptosis on the Metabolome in Cardiac Cells: The Role of Glutaminolysis"

_antioxidants, 2022, doi:10.3390/antiox11020278_

Round 1

Reviewer 1 Report

This paper elucidated the effects of ferroptosis on the metabolome in H9c2 cardioblasts by gas chromatography-mass spectrometry (GC-MS). The examination of the cellular metabolome indicated that a range of metabolic pathways, including the metabolism of amino acids and FA, as well as mitochondrial bioenergetics and glutaminolysis, are involved in ferroptotic signaling in H9c2 cardiomyocytes. The paper may serve as an interesting read for the journal’s audience. However, the following points need to be made clear and responded  with appropriate revisions for the manuscript to be considered for publication:

Originality and significance: There is literature available on the role of glutaminolysis in ferroptosis. In addition, the manuscript did not provide any mechanistic insights in this paper. It would be important to expand the introduction and discussion to show what additional information this paper can provide to advance thinking in the field.

Clarity and context: Though it is already there, but the discussion would be worth extending with relevant previous literature to make this study an engaging read. It is believed that the fatty acid metabolism has already been in the literature and a detailed comparison of the current results would be informative.

Validity: The authors provide an analysis of the metabolome but do not provide flux through the relevant pathways. Interestingly, LDH activity in RSL3-challenged cells was significantly increased. It will be very important and interesting to see the metabolic flux analysis in relevant pathways for ferroptosis induced by RSL3.

Methodology and data analysis: It is not clear what parameters were used for the analysis using MetaboAnalyst.ca . Extending the method section with information on analysis using MetaboAnalyst.ca would be useful for reproducibility.

Results and conclusions: It is very important and informative to conduct the metabolic flux analysis using the carbon-13 tracer for a comprehensive view of the metabolism and flux through the pathways identified using metabolome analysis.

Language: There are typos and grammatical errors in the language. It is recommended to proofread the manuscript and fix the language issues wherever applicable.

References: Use of references seems appropriate.

Author Response

Responses are attached.

Reviewer 2 Report

Antioxidants; antioxidants-1527022

Title: Effects of ferroptosis on the metabolome in cardiac cells: the role of glutaminolysis

Rodriguez-Graciani and co-authors conducted a fundamental study on the anti-ferroptotic effects of a novel reactive oxygen species (ROS) scavenger as TSM-1005-44 alongside ferrostatin-1 (ferroptosis inhibitor) and XJB-5-131 (mitochondrial-targeted ROS scavenger). In brief, the authors treated H9c2 cardioblasts with a glutathione peroxidase 4 inhibitor (RSL3) and collected biochemical and metabolomics data surrounding mitochondrial respiration and cellular viability. As a result of their findings, the authors conclude that the novel TSM agent is just as effective at countering ferroptosis relative to the structurally different ferrostatin and XJB agents. I have comments for further clarification of the study/manuscript below.

(1) Per the components of the ROS pathway (superoxide, peroxide, hydroxyl radicals), the known/proposed mechanism of action for RSL3 and various antioxidants should be clarified in the manuscript. Are various antioxidant agents more like superoxide dismutase or GPX/catalase in mechanism? Both? Are reactive nitrogen species (e.g., peroxynitrite) involved?

(2) As somewhat intertwined with comment #1, the reader should also know why GPX4 inhibition alone causes ferroptosis when there is the potential for one of the other eight GPX isoforms and/or catalase to compensate. Does activating/blocking multiple isoforms at once cause more or less ferroptosis?

(3) Also, did the authors try examining what TSM and XJB do to the cardioblasts in the absence of RSL3? For during RSL3 treatments, were antioxidants also tested in various combinations (e.g., TSM + XJB, TSM + XJB + ferrostatin)? The purpose of such speculation would give the readers a better idea as to the therapeutic potential of such agents used alone vs. in combination.

(4) As a potential study limitation, it would be ideal if the authors could speculate on whether the cultured cell line used is generally on par with freshly isolated cardiac cells (rodents or humans) with regard to metabolomic profiles and extent of glycolysis vs. mitochondrial respiration. Again, this is to help provide considerations for translational value if at least somewhat known. Obviously, some interventions (e.g., siRNA) require culture conditions outside of the authors’ experimental control.

(5) As perhaps another study limitation, the authors focus on an output of cell death. Does a dysregulation of oxidants vs. antioxidants always lead to cell death in a pathological scenario? What about more nuanced disturbances in cardiac calcium and electrical signaling to cause arrhythmias and/or growth factor signaling for hypertrophy? Inflammation?  

(6) Finally, the manuscript needs another thorough editing for fixing various typos. As one example, I see plenty of switching between use of “cystine” (i.e., two oxidized cysteines joined via a disulfide bond) and “cysteine”. Ensure that terms are indeed correct throughout.

Author Response

Responses are attached.
